# Perceptions of Adulthood and Mental Health

**DOI:** 10.3390/ijerph21060773

**Published:** 2024-06-14

**Authors:** Mediss Tavakkoli, Erick Valarezo, Luis F. García

**Affiliations:** 1Department of Biological and Health Psychology, Universidad Autónoma de Madrid, 28049 Madrid, Spain; maryamtavakoli86@gmail.com; 2Department of Psychology, Universidad Técnica Particular de Loja, Loja 110107, Ecuador; egvalarezo@utpl.edu.ec; 3Lleida Institute for Biomedical Research Dr. Pifarré Foundation (IRBLleida), 25198 Lleida, Catalonia, Spain

**Keywords:** perceptions of adulthood, subjective adult status, mental health, well-being, satisfaction with life, pathological personality

## Abstract

Background: In contrast to conventional definitions, the contemporary conceptualization of adulthood emphasizes psychological characteristics over sociodemographic milestones. At the same time, an increasing number of theorists propose that the way individuals view adulthood may have a significant impact on the mental health of both adolescents and adults. However, empirical examination of this hypothesis has been notably limited to date. The aim of this study is to explore the association between individuals’ perceptions of adulthood and multiple dimensions of mental health. Method: This study applied some adulthood markers and multiple mental health indexes (including well-being, optimism, Alexithymia, satisfaction with life, Goldberg’s index of mental health, the dark triad, and dimensional personality disorders) to a community sample comprising 1772 individuals in Spain, spanning ages from 16 to 93 years. Results: The findings support the overarching hypothesis, as perceptions of adulthood display strong correlations with nearly every assessed index of mental health, particularly those that comprise a dimension of negative emotions. These associations persist even after accounting for age and socio-economic status, and in alignment with the psychological paradigm of adulthood, they show a notable consistency across various age groups. Conclusions: This study establishes that such perceptions of adulthood represent a modifiable factor contributing to positive mental health. The implications of these findings for the formulation of public policies aimed at promoting mental health in the context of adulthood, as well as a number of future studies, are deliberated.

## 1. Introduction

The American Association of Psychology (APA) defines adulthood as the stage of human development characterized by complete physical growth, maturity, and various biological, cognitive, social, personality, and aging-related changes. This stage marks the culmination of brain maturation and the acquisition of personal and social skills essential for attaining status in society, engaging in mating and reproduction, and establishing social relationships in general [1]. While the existence of adulthood is universally acknowledged across societies, considerable debate surrounds the defining characteristics of an adult. This uncertainty is particularly pronounced when distinguishing between teenagers and adults. Cultural perceptions vary significantly regarding the traits and behaviors that qualify an individual as an adult [2,3,4]. Moreover, notable intra-cultural variations exist, as the very concept of adulthood evolves across generational cohorts and divergent sociodemographic groups [5,6,7,8,9].

Given the diverse perspectives on adulthood across cultures and social groups, accurately capturing the societal and individual constructs of adulthood is crucial. Initially, adulthood was marked by traditional and socially normative milestones, such as completing education, embarking on a career, marrying, and parenting. Indeed, the very definition of being an adult is profoundly shaped by the contextual significance attributed to these adult roles [10]. However, it has been argued that these age-specific milestones alone are insufficient to encapsulate the developmental essence and status of adulthood [11,12,13]. From this perspective, the characteristics most commonly associated with adulthood are psychological rather than sociodemographic, highlighting the importance of psychological development throughout adulthood, which extends beyond traditional societal roles.

Adopting this perspective, it has been asserted that the transition to adulthood is characterized more by psychological nuances and, notably, individualistic considerations [14]. The pathways young people take are diverse and non-linear, challenging the notion of a straightforward progression through life [15,16,17]. Instead of being solely marked by key social milestones, transitions to adulthood are shown to be varied, reversible, complex, and sometimes contradictory [18]. This transition is delineated by distinct psychological characteristics as evidenced by the fact that individuals at the same life stage may not uniformly identify with being an adult. These psychological characteristics have been identified as “perceptions of adulthood” in the literature [14,19,20].

Various theories have been proposed to elucidate the perceptions of adulthood, along with different age ranges to encapsulate the general characteristics of specific life stages. An early phase transitioning from adolescence to adulthood has been identified as “emerging adulthood”, spanning ages 18 to 29. Arnett describes this stage as one of identity exploration, instability, self-focus, open possibilities, and a feeling of being in limbo between adolescence and full-fledged adulthood [21]. These characteristics align closely with the transition to adulthood within a given cultural context, aiding in the delineation of the process and duration required to assimilate into long-term adult roles. This delineation moves beyond narrowly defined stages that fail to capture individuals’ personal perceptions of adulthood. Furthermore, Arnett has highlighted independence and autonomy as crucial elements for accurately understanding the transition from adolescence and achieving a successful adulthood [21].

Given its emphasis on the pivotal transition from adolescence to adulthood, the stage of emerging adulthood has garnered considerable attention. However, subsequent stages in the development of adulthood have also been proposed and explored by other scholars. Following emerging adulthood, the phase of established adulthood is suggested to span ages 30–45 [22], succeeded by midlife, which covers ages 46–59 [23], and culminating in older adulthood, beginning at age 60 and beyond [24]. A psychologically centered approach has been consistently applied to describe successful adaptation during these specific age stages. For instance, key factors identified for perceiving successful aging include health and independent functioning, psychological aspects such as cognitive, emotional, and motivational functioning, engagement in social roles and activities, financial and living situations, and the quality of social relationships [25,26].

A thorough review of the literature reveals a range of psychological indicators that signify a successful transition into adulthood [8]. These encompass physical health, psychological and emotional wellness, life skills, ethical conduct, the maintenance of healthy family and social relationships, educational attainment, constructive engagement in educational and occupational activities, and civic involvement. Notably, the majority of these aspects are psychological, highlighting the significance of an individual’s ability to integrate effectively within society (for example, through emotional well-being and the quality of relationships) and to be productive (evidenced by educational achievements and the cultivation of prosocial behaviors). 

### Mental Health and Adulthood

The World Health Organization’s definition of mental health as a state of well-being where an individual is able to recognize their abilities, handle life’s stresses, work productively, and contribute to their community underscores the importance of adaptability and productivity as hallmarks of sound mental health. This perspective closely aligns with the components of successful adulthood, including psychological and emotional well-being, life skills, and social integration, as previously discussed.

Despite the clear theoretical connections between the psychological meaning of successful adulthood and mental health, empirical research exploring this relationship has been limited. Sharon [13] investigated the link between markers of adulthood, as initially defined by Arnett [19], and measures of social well-being and self-esteem in a study predominantly involving young women. The findings revealed that the markers of adulthood explained 3.7% and 3.5% of the variance in social well-being and 4% and 2% of the variance in self-esteem for males and females, respectively. Remarkably, the predictive power was almost entirely attributable to the relational maturity scale, highlighting the significance of forming healthy relationships. Furthermore, other authors found a correlation of 0.27 between a sense of adulthood and well-being in a study with older adult cancer survivors [27]. Nevertheless, the decision to use just one item for assessing adulthood perception in this research could have resulted in an undervaluation of the true relationship between the sense of adulthood and overall well-being.

Previous research that focuses solely on well-being as the singular metric [13,27] fails to fully capture the extensive spectrum of mental health. It is important to note that subsequent authors have expanded this perspective by including a diverse array of disorders, such as anxiety, disinhibition/impulsivity, stress-related, substance use/addiction, psychotic, personality, and mood disorders, among others. This broader approach provides a more comprehensive representation of mental health [28,29].

This approach underlines the necessity of exploring the relationship between perceptions of adulthood and mental health through a multifaceted lens that encompasses both psychopathological and social well-being aspects [30,31], as well as physical health [32].

Mental health is shaped by a multitude of factors, including personality traits, social relationships, sleep quality, social attainment and identification, socio-economic status, and physical health [33,34,35,36,37]. Perceptions of adulthood may represent another modifiable factor that contributes to mental health during adulthood and later stages of life, yet this potential remains largely unexplored. Given that individuals’ self-perceptions as adults can influence their mental well-being, it is imperative to investigate the link between perceptions of adulthood and mental health [24].

The objective of this study is to explore the association between perceptions of adulthood and various dimensions of mental health. By extending the existing body of literature, this study aims to highlight the potential role of adulthood perceptions in enhancing our understanding of mental health. If significant associations are discovered, it would suggest that fostering a perception of adulthood and achieving adult status could positively impact the mental health of individuals, communities, and society at large.

## 2. Materials and Methods

### 2.1. Participants

The study’s participants comprised 1772 individuals (754 males and 1018 females) from the Spanish general population, with an average age of 39.58 years (SD = 20.04), where males had an average age of 41.90 years (SD = 19.63) and females had an average age of 37.87 years (SD = 20.17). The age distribution ranged from 16 to 93 years. 

Consistent with the prior literature [24], the sample was categorized into four age groups to align with the developmental stages of adulthood: (1) emerging adulthood (aged 18–29 years), which included 788 participants (44.5% of the sample; 290 males [36.8%] and 498 females [63.2%]); (2) established adulthood (aged 30–45 years), which comprised 116 participants (6.5%; 57 males [49.1%] and 59 females [50.9%]); (3) midlife (aged 46–59 years), which included 623 participants (35.2%; 296 males [47.5%] and 327 females [52.5%]); and (4) older adulthood (aged 60 years and above), which included 245 participants (13.8%; 111 males [45.3%] and 134 females [54.7%]).

To assess the life milestones achieved by individuals in terms of social position, this study utilized Hollingshead’s Social Position Index (SPI) [38,39]. The SPI is derived from two 7-point scales: an Occupation Scale, ranging from 1 (higher executives) to 7 (unskilled employees), and an Education Scale, ranging from 1 (graduate professionals) to 7 (less than seven years of school). The SPI score is calculated using the formula [SPI = (Occupation score × 7) + (Education score × 4)], where lower scores indicate a higher social position. According to the scoring range provided by Hollingshead and Redlich [39], the distribution of social positions within the present sample is as follows: upper (<17) includes 973 participants (54.9%), upper-middle (17–31) includes 434 participants (24.5%), middle (32–47) includes 215 participants (12.1%), low-middle (48–63) includes 105 participants (5.9%), and low (>63) includes 45 participants (2.5%). This method of quantifying social position provides a comprehensive understanding of the participants’ social statuses, allowing this study to explore the role of social position in the relationship between perceptions of adulthood and mental health outcomes. Note that some necessary information, such as marital status, required to compute the four-factor version of the SPI was not collected; therefore, we used the two-factor version. 

### 2.2. Instruments

#### 2.2.1. Perceptions of Adulthood

Two distinct tools were employed to evaluate perceptions of adulthood: the Revised Markers of Adulthood scale [20] and the five-item measure of subjective adult status devised by Wright and Von Stumm [24]. These instruments are key to deciphering the contemporary transition into adulthood and individuals’ personal sentiments regarding their status as adults.

*Markers of Adulthood* [20]: The original Markers of Adulthood scale [19] underwent refinement by Norman and associates [20], providing a more detailed perspective on adulthood. This revised scale enables participants to evaluate the significance and their accomplishment of particular markers of adulthood, covering aspects like independence, legal benchmarks, role transitions, and relative maturity. The revised scale comprises 22 items, yielding two distinct evaluations: importance and achievement. Initially, individuals assess the importance of each marker for becoming an adult on a scale ranging from nothing (1), a little (2), quite (3), to very important (4). Subsequently, they gauge their attainment of these markers using a three-point scale: no (1), something (2), and yes (3). These items collectively evaluate four dimensions: independence (e.g., “Being financially independent from parents”) with 8 items; legality markers (e.g., “Having reached the age of legal adulthood”) with 4 items; role transitions (e.g., “Being married”) with 5 items; and relative maturity (e.g., “Accepting responsibility for the consequences of your actions”), also with 5 items [20]. The results of the adaptation process to the Spanish language and cultural context demonstrate adequate reliability for the scale, with alpha coefficients of 0.87 and 0.95 for the importance and achievement overall scores, respectively. Cronbach’s alpha values ranged from 0.67 to 0.85 for the importance scales and from 0.64 to 0.96 for the achievement scales. A similar factor structure to the original American version was also found [40].

*Subjective adult status* [24]: Wright and Von Stumm introduced a five-item measure for assessing subjective adult status, aiming to minimize measurement errors associated with previous single-item approaches [24]. This refined measure encompasses the following statements: “I feel like an adult”, “I no longer feel like a child”, “Other people consider me an adult”, “Other people treat me like a child” (which is reverse-scored), and “I think of myself as a grown-up person”. Participants evaluated these items using a 5-point Likert scale from “never” to “always”, with the scoring as follows: never = 1, rarely = 2, sometimes = 3, often = 4, and always = 5. A higher aggregate score suggests a more frequent self-perception of adulthood compared to a lower score. To adapt the scale to the Spanish language, the items were first translated by a Spanish psychology lecturer fluent in English. Subsequently, another Spanish psychology lecturer, also fluent in English, performed a back-translation. The back-translated version matched perfectly with the original English version. In the Spanish context, the reliability was indicated by a Cronbach’s alpha of 0.64. This coefficient could increase to 0.77 (closer to the original version’s 0.82 [24]) if item 2 (“I no longer feel like a child”) was removed, and to 0.84 if item 4 (“Other people treat me like a child”) was also excluded. However, the original five-item composition has been retained for comparative purposes.

#### 2.2.2. Mental Health Indexes

This study utilizes a comprehensive set of well-established instruments that collectively provide a broad evaluation of participants’ mental health: 

*1. Well-Being (WB):* Using the Spanish adaptation of an 18-item measure that evaluates six dimensions of well-being—autonomy, environmental mastery, personal growth, positive relations, purpose in life, and self-acceptance [41,42]—this utilizes a seven-option Likert-type response format that ranges from “totally agree” (1) to “totally disagree” (7). For the present study, the total scale’s internal consistency is high (Cronbach’s alpha = 0.84), indicating reliable measurement.

*2. Satisfaction with Life (SL):* Measured by the Spanish version of the Satisfaction with Life Scale consisting of 5 items with a Likert-type answer format from totally disagree to totally agree [43], the scale shows good reliability in the present sample (Cronbach’s alpha = 0.80).

*3. Optimism (OP):* The Spanish adaptation of the Life Orientation Test includes 10 items scored on a four-point Likert scale from totally disagree to totally agree [44]. It exhibits satisfactory internal consistency (Cronbach’s alpha = 0.73).

*4. Self-Esteem (SE)*: The Spanish adaptation of the Rosenberg Self-Esteem Scale is utilized [45], consisting of 10 items designed to measure self-esteem. It features a balanced mix of five positively and five negatively worded items, with responses captured on a 4-point scale. This scale is noted for its excellent reliability, evidenced by a Cronbach’s alpha of 0.88.

*5. Alexithymia (AL):* The TAS-20 is a 20-item self-report measure that assesses Alexithymia on a 5-point Likert scale, where responses range from 1 (strongly disagree) to 5 (strongly agree). Alexithymia is characterized by a difficulty in identifying and expressing emotions. For this study, a validated Spanish version of the scale was employed [46], which has demonstrated high reliability, with a Cronbach’s alpha of 0.85.

*6. General Health Questionnaire (GHQ):* General health is evaluated using the GHQ-28, which comprises 28 items spanning four subscales, utilizing a Likert-type response format [47]. Scores vary from 0 (better than usual) to 3 (much worse than usual), indicating the range of perceived changes in condition. The overall scale is noted for its very high reliability (Cronbach’s alpha = 0.94).

*7. Social Well-being:* We have utilized the Spanish adaptation of the social well-being scale [48], which assess five dimensions: social integration, social acceptance, social contribution, social actualization, and social coherence. Items are assessed using a scale that ranges from 1 (strongly disagree) to 5 (strongly agree). The Cronbach’s alpha values for these scales fall between 0.69 and 0.87, demonstrating satisfactory reliability.

*8. Dark Triad (The Dirty Dozen):* The Spanish version of the Dirty Dozen measures the dark triad personality traits of Machiavellianism, Psychopathy, and Narcissism [49]. The 12-item measure uses a 7-point Likert scale from strongly disagree to strongly agree, with internal consistency alphas of 0.81 for Machiavellianism, 0.71 for Psychopathy, and 0.85 for Narcissism, reflecting good reliability for assessing these traits.

*9. Dimensional Personality Disorders (The Personality Inventory for ICD-11–PiCD):* The Spanish adaptation of the PiCD assesses five domains of the ICD-11 dimensional model of personality disorders: negative affectivity, detachment, dissociality, disinhibition, and anankastia [50]. It consists of 60 items, 12 for each domain, with responses on a 5-point Likert scale from strongly disagree to strongly agree. The domains’ Cronbach’s alpha ranged from 0.78 to 0.84, indicating good internal consistency in the present sample. 

### 2.3. Procedure

Undergraduate psychology students received training in the application of psychological assessment tools. As part of a routine exercise, they were tasked with administering a set of psychological measures in a paper–pencil format, including those discussed in this study, to seven individuals: the students themselves, one male and one female aged between 18 and 30 years, one male and one female aged between 31 and 50 years, and one male and one female over the age of 51 years. This participant selection strategy was designed to capture a wide range of ages and genders. To ensure confidentiality, no names, personal identification numbers (such as identity cards), or any other personal details were recorded. However, all participants were informed about the study guidelines approved by the university’s ethical committees. To enhance motivation among participants, scores from the NEO-PI-R personality trait assessment were shared with all individuals involved. Each university student was assigned a unique, random code. Results were then provided anonymously, showcasing only the participant’s code, alongside the age and sex of the individual. Consequently, each student was able to recognize their own responses, as well as those of the specific individuals they assessed, using the provided demographic information to aid in identification. 

We would like to emphasize that all participants were informed that anonymous data could be used for research purposes. A database of all information about this study was saved in a computer protected by a personal password. The handling of the information was carried out in accordance with the confidentiality rules set out in the Spanish Organic Law 3/2018 on Data Protection and Guarantee of Digital Rights, Helsinki Declaration, in the Council of Europe Convention on Human Rights. 

### 2.4. Analysis 

The envisioned analysis strategy adeptly employs statistical techniques to delve into the relationships among markers of adulthood, subjective adult status, and an extensive spectrum of mental health indexes. Initially, by computing correlations between total scores, the four scales of importance and achievement of markers of adulthood, subjective adult status, and all mental health indexes, it is possible to pinpoint particular association patterns among these constructs both within the overall sample and throughout various stages of adulthood.

Secondly, given the extensive array of mental health scales and their diversity, they underwent simplification through the factor analysis multivariate method. This approach is viable because mental health indexes are known to share common variances [29,51]. The extraction technique employed was be principal component analysis, and to ensure maximum differentiation between factors, an orthogonal rotation (Varimax) was applied. Considering the varied nature of the mental health indexes, this study does not predefine a theory on the number of factors to extract. However, the PiCD has consistently revealed a replicable four-factor structure across different cultures [29,52,53]. Thus, an initial extraction of four factors was pursued. Factor scores were then determined using the regression method. This factor analysis was carried out exclusively on the entire sample.

Finally, conducting hierarchical linear regression analyses is a suggested method for dissecting the contributions to mental health factors. By methodically incorporating varying sets of predictors into the analysis, this approach will make it possible to isolate the distinct variance that each group of variables contributes to explaining the variance of the mental health factors identified through the preceding factor analysis. This structured methodology enables a clearer understanding of how different predictors individually and collectively influence mental health factors. Here is a breakdown of how the regression analysis steps were organized and their implications:

Step 1—Demographics: Entering age and the SPI first allows you to account for basic demographic influences on mental health. This step establishes a foundational understanding of how these fundamental variables relate to mental health factors.

Step 2—Importance of Adulthood Markers: Adding the scales related to the perceived importance of adulthood markers next helps to assess how much additional variance in mental health outcomes can be explained by individuals’ perceptions of what it means to be an adult. 

Step 3—Achievement of Adulthood Markers: This phase of the analysis plays a crucial role in elucidating the impact that achieving adult roles, milestones, and responsibilities has on mental well-being. It provides valuable insights into the broader effects of perceptions of adult development on psychological health.

Documenting the shifts in R² (coefficient of determination) at each stage offers a quantifiable measure of the contribution each set of variables makes towards explaining the variance in mental health factors. This approach not only quantifies the impact of different predictors on mental health but also clarifies the extent to which each factor enhances our understanding of these outcomes. Additionally, the standardized beta coefficients in the final model will highlight the relative importance of each predictor when all other variables are accounted for, offering insights into the most influential factors. Performing this analysis for the total sample and separately for each of the four adulthood stages allows for a nuanced understanding of how these relationships vary across different stages of adulthood. This method offers a tool to highlight key stages at which perceptions and achievements concerning adulthood exert a significant influence on mental health.

## 3. Results

The findings outlined in Table 1 indicate the Pearson correlations among the importance and achievement rates of markers of adulthood, subjective adult status, and all mental health indexes. A coherent pattern of correlation is observed between achievement scales and subjective adult status, aligning with favorable mental health outcomes. These include well-being, satisfaction with life, optimism, self-esteem, general health, and negative affectivity, along with various dimensional personality disorders and dark triad traits. This relationship underscores the importance of accomplishment and self-perception in adulthood in promoting psychological well-being and mitigating negative psychological traits. This implies that achieving adulthood markers and having a strong subjective sense of being an adult are linked to better mental health. Correlations between the importance scales and mental health indexes are generally lower and often close to zero. This indicates that the ratings of importance of adulthood markers might not significantly impact mental health outcomes, underscoring the greater relevance of perception of actual achievement. 

While correlations tend to be higher in established adulthood and midlife compared to emerging and older adulthood, the overarching pattern of stronger associations with achievement scales and subjective adult status remains consistent across age stages. This consistency across life stages reinforces the idea that the perception of achieving markers of adulthood is related to better mental health outcomes, although the strength of these relationships may vary with age.

The factor analysis conducted to streamline the mental health indexes into a smaller set of common psychological characteristics produced compelling results that support a conceptual framework (Table 2). The Kaiser–Meyer–Olkin (KMO) measure indicating good sampling adequacy (0.837) and the highly significant Bartlett’s test of sphericity (11,959.726; degrees of freedom = 171, *p* < 0.001) suggest that the dataset was suitable for factor analysis. Factor 1 (negative emotions) comprises negative affectivity and scales related to negative and positive emotions. This factor, inversely associated with the presence of negative emotions, underscores the psychological dimension that captures the tendency towards experiencing adverse emotional states. Factor 2 (social well-being) includes social well-being scales, with the detachment scale loading negatively. This factor highlights the importance of social integration and acceptance in psychological well-being and inversely associates with feelings of detachment. Factor 3 (antisocial) consists of measures linked to antisocial behavior, including the dark triad and dissociality. This factor encapsulates traits and behaviors often associated with antisocial tendencies. Finally, Factor 4 (disinhibition vs. anankastia) encompasses disinhibition and anankastia at opposite ends, reflecting the literature-supported dichotomy between impulsivity/disinhibition and compulsivity/orderliness [29,52]. 

The identified factors align well with theoretical expectations and previous research, providing a nuanced understanding of the multidimensional nature of disorders and mental health [51,53]. These factors offer a structured way to examine how different aspects of psychological functioning relate to markers of adulthood and subjective adult status. The retention of an orthogonal solution is justified by the general independence of the factors, except for the theoretically anticipated correlation between the third and fourth factors (0.365) [54]. The low correlations among most factors (0.128 being the highest apart from the aforementioned exception) further support the decision for an orthogonal rotation, indicating distinct, largely independent underlying dimensions of mental health.

Table 3 reveals stronger correlations between achievement scales, subjective adult status, and especially the negative emotions factor, with a lesser correlation observed with the antisocial factor. Associations with negative emotions remain consistent across all age stages, while social well-being correlates with achievement scales during established adulthood and midlife, and the antisocial factor correlates with achievement scales in the total sample but not within specific age stages.

Table 4 demonstrates that achievement scales serve as key predictors in hierarchical regressions, contributing up to an additional 12% of the variance in the negative emotion factor for the total sample, after accounting for age, SPI, and achievement scales. In the four age stages, this increment ranges from 6% in older adulthood to 26% in established adulthood. The relative maturity scale emerges as the strongest predictor of negative emotions, consistently showing the low predictive power of age, SPI, and importance scales across age stages. Notably, the percentage of variance explained by the other three factors is lower, particularly for the antisocial and disinhibition factors.

## 4. Discussion

The present manuscript supports the notion that an individual’s perception of being integrated and productive within society plays a significant role in their mental health. In detail, perceptions of adulthood account for about 10% of the reliable variance in different mental health measures, highlighting their significance in addressing individual and societal mental health differences. This insight is particularly valuable in psychological research aimed at pinpointing and comprehending the factors influencing mental health, which is essential for creating effective interventions and support mechanisms. The finding that achievement scales, rather than importance scales, predict mental health outcomes suggests that the actual perception of achieving adulthood milestones, particularly psychological ones, is most beneficial. It is not the perceived importance of these elements, but rather the perception of having attained them, that positively influences mental health. This distinction highlights the practical implications of adulthood achievements, suggesting that the perception of realizing key developmental tasks is more impactful on an individual’s mental health than simply valuing these milestones.

The pivotal role of achieving relational maturity as the most predictive factor for the connection between perceptions of adulthood and mental health underscores a significant finding in this study, replicating previous studies [13,27]. This suggests that a mature understanding and management of one’s abilities, alongside a willingness to accept the consequences of one’s actions, are crucial elements linking adulthood perceptions to mental health outcomes. These insights highlight the critical role of psychological maturity in adulthood and mental well-being, reinforcing the importance of this concept as outlined in reference [20].

Overall, the findings indicate that the essence of transitioning to and thriving in adulthood is more deeply connected to psychological perceptions than to sociological accomplishments. This distinction is particularly notable in that the scales most associated with mental health outcomes focus on individual psychological development rather than sociological milestones such as legacy markers or role transitions. This observation reinforces the idea that mental health in the context of adulthood is deeply intertwined with personal growth and self-perception. In addition, the limited predictive power of age [55] and social position index on mental health further supports the primacy of psychological over sociological factors in determining well-being during adulthood. 

Interestingly, the variance in the strength of associations across the four adulthood stages indicates that the transition to and experience of adulthood are not monolithic but vary significantly with age. This insight introduces a refined perspective on adulthood, emphasizing the varying impact it has across different life stages. The discovery that perceptions of adulthood are most predictive during established adulthood (ages 30–45) suggests that this stage’s unique social and psychological pressures heavily influence mental health. This stage of adulthood is often marked by intensified demands from various aspects of personal and professional life, which may amplify the impact of adulthood perceptions on mental health.

The current study underscores the value of utilizing perceptions of adulthood instruments in mental health research. These tools enhance our understanding of how adulthood is perceived and realized across diverse individuals, highlighting the intricate and multidimensional process of adult identity development. Notably, the subjective adult status scale [24], with just five items, has shown similar associations with mental health indices as the more comprehensive markers of adulthood. Specifically, this brief scale presents comparable correlations with negative emotions and, in some age stages, with social well-being. This pattern reinforces the conclusion that the psychological perception of being an adult is crucial to the relationships between perceptions of adulthood and mental health, contributing significantly to positive mental health outcomes. Finally, from an applied standpoint, given its brevity, simplicity, and quick application, the subjective adult status scale emerges as a promising tool for epidemiological studies or research scenarios where time constraints limit the use of extensive psychological instruments.

### 4.1. Future Studies 

As a groundbreaking study in this field, it lays the foundation for numerous future research avenues. Surprisingly, perceptions of adulthood had little impact on social well-being, despite healthy social relationships being a key aspect of adulthood [8]. This may be due to the tools used not specifically assessing social aspects of adulthood, focusing instead on individual self-perception without much reference to societal or communal contexts. The present study examines two measures of perception of adulthood, but they do not encompass the full range of possibilities. A recent framework, the CARES taxonomy, suggests a broader spectrum of qualities, including cognitive maturity, sense of aging, self-reliance, Eudaimonia, and social convoy [24]. It is plausible that the social convoy aspect, which pertains to the network of relationships in a person’s life, might have a stronger connection with social well-being [24]. Therefore, future research should explore the relationship between social well-being indexes and the CARES taxonomy or other comprehensive frameworks of perceptions of adulthood. 

The absence of direct measures of anxiety, depression, and other internalizing disorders in the current study highlights an essential area for future research. Considering the centrality of these concepts in any comprehensive mental health model [51], their inclusion could significantly enhance our understanding of the nuanced relationship between perceptions of adulthood and mental health outcomes. Previous studies showed a linkage between perceptions of adulthood and anxiety and depression [27,56] that is theoretically consistent with our findings related to the negative emotions factor. Given that this factor encompasses aspects theoretically aligned with anxiety and mood disorders [57], integrating direct measures of these conditions could provide a more detailed understanding of the psychological challenges associated with the transition to and experiences of adulthood. 

Moreover, broadening the scope of this research to encompass other aspects of mental health, such as substance abuse, could provide a more comprehensive understanding of the challenges and vulnerabilities associated with adulthood. For instance, substance abuse, often linked to impulsivity or coping mechanisms for dealing with stress, anxiety, and depression [57], could further elucidate the complex interplay between mental health and the societal and psychological aspects of adulthood. Incorporating these additional mental health indexes in future studies would not only validate and extend the findings of the present research but also contribute to a deeper and more comprehensive understanding of adulthood’s psychological impact.

Adulthood entails biological brain maturation [1], with notable disparities between chronological and brain age [58]. Investigating whether differences in brain age are linked to perceptions of adulthood and exploring whether these perceptions mediate the potential impact of mismatches between chronological and brain age, or other biological markers, on behavioral outcomes [59] would be intriguing avenues for future research. 

Recognizing the present paper as a descriptive first step, there is a need for a deeper exploration of the mechanisms underlying these associations. Implementing longitudinal research designs is crucial for untangling the cause-and-effect relationships between perceptions of adulthood and various psychological, biological, and social variables. Such studies would provide invaluable insights into how changes in the perception of adulthood over time impact mental health and vice versa, capturing the dynamic nature of these constructs throughout the lifespan. Acknowledging adulthood as a multidimensional construct necessitates the integration of various related concepts to develop a comprehensive understanding of how the sense of adulthood evolves. This approach will help elucidate the interplay between subjective adulthood and other aspects of an individual’s mental health, life, and identity. For instance, given the significant impact of personality traits on well-being [60], supported by meta-analytic evidence [34], incorporating personality into studies on adulthood perceptions and mental health is essential. Understanding how perceptions of adulthood interact with personality traits, as well as with other factors affecting mental health [61,62], could reveal key pathways for enhancing mental health outcomes.

To address the variability in adulthood perceptions across cultures [2,3,4,10], future research should aim to replicate our study in countries with different sociological frameworks, particularly in non-Western countries. This expansion would provide valuable insights into the cultural nuances of adulthood and how these differences influence mental health outcomes. Cultural norms, values, and expectations significantly shape the transition to adulthood, making it imperative to examine these constructs in varied cultural settings. For instance, it has been described that Spanish people emphasize psychological aspects (emotional and sexual) over financial or chronological criteria in the transition to adulthood compared to other countries [10]. They are also characterized by a longer stay in the parental home, higher rates of enrollment in higher education, delayed workforce entry, and older marriage ages. Different results could be found in countries with different sociological profiles. Alongside cultural diversity, the inclusion of diverse social groups within the same cultural context is crucial. This approach allows for an examination of how socio-economic status, education levels, and other social determinants impact perceptions of adulthood and their association with mental health outcomes. Conducting comparative analyses between Western and non-Western countries, as well as across different social groups within the same country, would shed light on the relative impact of cultural and social factors on adulthood perceptions. Such analyses could highlight universal versus context-specific elements of the transition to adulthood and its relationship with mental health. Lastly, considering the somewhat low reliability of the subjective adult status scale observed in the Spanish context, it is important to replicate its psychometric properties in other countries to support the universal applicability of this promising scale.

The overrepresentation of university students and individuals from higher social positions, coupled with the limited number of participants from the established adulthood stage in the current sample, represents a notable limitation. This underscores the importance of including a more diverse range of social and socio-economic backgrounds in future studies to ensure a more comprehensive understanding of adulthood and its associations with mental health outcomes. Ensuring that future replication studies include more representative samples is key to enhancing the validity and applicability of the findings. Stratified sampling methods or purposive sampling strategies can help achieve greater diversity and representation, allowing for a more nuanced understanding of adulthood across different segments of the population.

### 4.2. Implications about Public Health

The implications of our study for personal and public health emphasize the significance of cultivating positive perceptions of adulthood and enhancing subjective adult status to promote long-term mental health. Although our research does not directly address interventions, the findings offer valuable insights into potential strategies for enhancing mental well-being through targeted support during the transition to adulthood. The robust association between coping with negative emotions and perceptions of adulthood suggests that psychological interventions aimed at alleviating anxiety and negative mood could be particularly advantageous. Techniques such as cognitive–behavioral therapy (CBT), mindfulness-based stress reduction (MBSR), and resilience training offer effective means for individuals to manage and mitigate the effects of negative emotions, thereby nurturing a healthier perception of adulthood and enhancing overall mental well-being. Additionally, programs aimed at reshaping attitudes toward adulthood could play a pivotal role in facilitating the transition for emerging adults [24]. By promoting a more adaptable and pragmatic view of adulthood, these programs have the potential to alleviate the sense of uncertainty often associated with this adulthood stage. Therefore, highlighting the significance of establishing stability in one or two key aspects of life [63] could offer a concrete objective for emerging adults. This approach recognizes the difficulties of simultaneously achieving numerous adulthood milestones and proposes that concentrating efforts on attaining stability in specific domains (e.g., career, relationships) can greatly enhance a sense of adult identity and well-being.

Public health initiatives have the potential to raise awareness about the influence of perceptions of adulthood on mental health and offer resources and assistance for individuals navigating this transition. Educational workshops, mentorship programs, and media campaigns could serve as effective platforms for disseminating these messages and promoting a positive perspective on adulthood. Furthermore, these initiatives could underscore the significance of mental health care, advocate for seeking psychological support, and emphasize the accessibility of community resources to aid individuals in addressing the challenges of adulthood. Policymakers could leverage these insights to shape policies aimed at bolstering mental health during the transition to adulthood. This might entail allocating funding for mental health services tailored to young adults, backing education and employment initiatives that ease the transition to adulthood, and incorporating mental health education into school curricula to equip adolescents with the skills needed to navigate challenges effectively. By prioritizing interventions and strategies that address the psychological aspects of transitioning to adulthood and managing negative emotions, we have the potential not only to enhance individual well-being but also to contribute to broader public health outcomes. 

Achieving this requires a concerted effort involving collaboration among mental health professionals, educators, policymakers, and community leaders to ensure that emerging adults receive the necessary support to flourish. It is imperative to adopt a comprehensive approach that extends beyond psychological interventions to encompass social, economic, and educational measures, forming a holistic pathway towards improving perceptions of adulthood and enhancing overall mental health. This holistic strategy acknowledges the multifaceted nature of adulthood transitions and recognizes the diverse needs of individuals across various life stages. Therefore, the present framework advocates for participation in social programs and community activities that cultivate independence, adaptability, and capability. These initiatives may encompass volunteer opportunities, sports, arts, and other group activities [32,64] that not only foster social interaction but also reinforce individuals’ sense of belonging and identity within the adult community [61]. Conversely, interventions focused on increasing educational attainment and promoting sustained engagement in the educational system can profoundly influence perceptions of adulthood and mental health. By facilitating access to higher education and vocational training, individuals are better prepared to confront the challenges of adulthood, thereby enhancing their self-perception and societal contribution. This approach aligns with research indicating that higher levels of education are associated with a reduced risk of mortality [65]. 

Finally, the present paper recommends redirecting attention towards psychosocial interventions tailored specifically for individuals in the established adulthood stage. Customized programs aimed at assisting adults in managing the intricacies and obstacles of this stage, including aspects such as work–life balance, parenting, and career advancement, are crucial. Support groups, counseling services, and workplace initiatives can all play significant roles in mitigating the mental health risks associated with this life stage.

## 5. Conclusions

The present study supports the association between perceptions of adulthood and mental health, particularly in relation to the risk of experiencing negative emotions. The findings also suggest that psychological aspects of adulthood are key to this association, opening an intriguing framework for the prevention and treatment of mental health issues.

The association varies across different adult stages. Developing strategies specifically targeted at each age group can help mitigate the onset and progression of mental health issues, emphasizing the importance of early detection and intervention. The high prevalence of mental health issues among adolescents and young adults underscores the significance of focusing on these age stages as crucial periods for the onset of psychiatric disorders [66]. Additionally, the majority of psychiatric disorders manifest during these stages [28]. The transition to adulthood is a critical time for fostering positive growth and integration into society, providing a stable and nurturing environment for individuals to thrive as healthy, contributing members of the community. 

Understanding the risk and protective factors during these stages is essential for preventing mental health problems and improving treatment outcomes. Therefore, altering erroneous and maladaptive perceptions of adulthood and reinforcing subjective adult status could serve as valuable psychological tools to address these issues, potentially leading to long-term positive effects on mental health. The paradox concerning aging and subjective well-being, as discussed by Hansen and Blekesaune [67], suggests that despite the physical and social losses often associated with aging, subjective well-being tends to remain stable or even improve. This paradox could be partially attributed to enhanced perceptions of adulthood and social status serving as compensatory factors for the losses experienced during aging. Such perceptions may not only bolster well-being in later life but also potentially contribute to longevity [68]. Adopting a developmental perspective when addressing mental health issues holds promise for research and intervention. By considering the broader implications for well-being and longevity, these approaches could pave the way for more effective prevention strategies and support systems. These initiatives would not only directly target mental health issues but also enhance individuals’ abilities to navigate life’s challenges and transitions in a manner that fosters overall health and well-being. The present manuscript advocates for global public policies addressing various aspects—psychological, social, and educational—to assist individuals across different age stages in cultivating a healthy sense of adulthood and promoting holistic development. These policies hold the potential to tackle immediate mental health challenges during critical developmental stages while also safeguarding long-term mental health and potentially contributing to longevity. This highlights the interconnectedness of mental and physical health across the lifespan. Given that adulthood is a stage of ongoing growth and development unique to each individual, these policies should be as individualized as possible to cater to diverse needs and experiences.

## Figures and Tables

**Table 1 ijerph-21-00773-t001:** Correlations among perceptions of the importance (Imp) and achievement (Ach) of markers of adulthood and subjective adult status with mental health indexes in the total sample and the four age stages.

		TOTAL SAMPLE																		
		WB	SL	OP	SE	AL	GHQ	SWB-SI	SWB-SA	SWB-SC	SWB-SL	SWB-SH	MAC	PSY	NAR	NA	DT	DL	DN	AN
Imp	Total	0.01	0.10	0.08	0.09	−0.05	0.01	0.11	−0.04	0.08	−0.04	−0.07	−0.07	−0.05	0.02	−0.01	−0.08	0.00	−0.11	0.10
	Role	0.00	0.12	0.13	0.12	−0.01	−0.04	0.07	0.00	0.07	−0.09	−0.07	−0.08	−0.01	−0.02	−0.08	−0.04	0.02	−0.10	0.04
	Independence	0.00	0.04	0.03	0.05	−0.03	0.03	0.06	−0.08	0.02	−0.01	−0.05	−0.03	−0.03	0.05	0.03	−0.05	0.01	−0.08	0.10
	Legality	−0.04	0.05	0.04	0.03	0.02	0.04	0.08	−0.07	0.06	−0.04	−0.10	−0.03	0.01	0.03	0.04	−0.05	0.05	−0.03	0.06
	Maturity	0.08	0.07	0.03	0.02	−0.18	0.03	0.15	0.06	0.11	0.07	0.05	−0.08	−0.17	−0.01	0.02	−0.12	−0.10	−0.12	0.12
Ach	Total	0.14	0.14	**0.29**	**0.36**	**−0.28**	−0.17	0.12	**0.24**	0.18	0.11	0.01	**−0.27**	−0.17	**−0.29**	**−0.36**	−0.12	**−0.22**	**−0.37**	0.02
	Role	0.12	0.13	**0.25**	**0.29**	**−0.22**	−0.12	0.09	**0.21**	0.12	0.09	−0.03	**−0.26**	−0.17	**−0.27**	**−0.27**	−0.12	**−0.22**	**−0.32**	0.02
	Independence	0.10	0.10	**0.26**	**0.33**	**−0.24**	−0.16	0.07	**0.23**	0.15	0.10	−0.01	**−0.26**	−0.15	**−0.28**	**−0.34**	−0.08	**−0.22**	**−0.35**	0.01
	Legality	0.14	0.10	**0.25**	**0.31**	**−0.28**	−0.16	0.13	0.19	0.18	0.10	0.04	−0.17	−0.12	−0.19	**−0.30**	−0.12	−0.12	**−0.27**	-0.02
	Maturity	0.17	0.22	**0.25**	**0.34**	**−0.33**	−0.18	**0.24**	0.14	**0.25**	0.12	0.14	**−0.22**	−0.14	−0.19	**−0.32**	−0.14	−0.13	**−0.28**	0.09
Subjective adult status		0.18	0.15	**0.23**	**0.33**	**−0.25**	−0.19	**0.22**	0.18	**0.20**	0.12	0.05	**−0.24**	−0.14	−0.20	**−0.31**	−0.12	−0.18	**−0.34**	0.05
		**EMERGING ADULTHOOD**																		
		**WB**	**SL**	**OP**	**SE**	**AL**	**GHQ**	**SWB−SI**	**SWB−SA**	**SWB−SC**	**SWB−SL**	**SWB−SH**	**MAC**	**PSY**	**NAR**	**NA**	**DT**	**DL**	**DN**	**AN**
Imp	Total	0.03	0.06	0.04	0.05	−0.05	0.05	0.08	−0.09	0.02	−0.10	−0.08	−0.01	−0.02	0.08	0.08	−0.04	0.07	−0.02	0.06
	Role	0.03	0.10	0.09	0.10	−0.01	−0.02	0.07	−0.05	0.06	−0.17	−0.06	0.01	0.04	0.06	−0.03	−0.02	0.14	0.03	-0.03
	Independence	0.01	−0.02	−0.01	0.01	−0.01	0.05	0.02	−0.13	−0.06	−0.05	−0.06	0.02	0.01	0.09	0.12	0.01	0.05	−0.02	0.09
	Legality	0.00	0.05	0.03	0.04	0.00	0.05	0.07	−0.09	0.02	−0.06	−0.13	0.02	0.03	0.05	0.09	−0.02	0.08	0.01	0.03
	Maturity	0.07	0.07	0.00	−0.02	−0.16	0.07	0.10	0.06	0.07	0.08	0.06	−0.12	**−0.21**	0.00	0.07	−0.10	−0.13	−0.11	0.12
Ach	Total	0.05	0.08	0.10	0.22	**−0.26**	−0.10	0.15	0.05	**0.22**	−0.04	0.07	−0.04	−0.03	−0.07	−0.19	−0.09	0.07	−0.13	0.01
	Role	0.03	0.07	0.03	0.08	−0.15	−0.01	0.11	−0.02	0.11	−0.04	0.01	−0.06	−0.05	−0.01	0.00	−0.15	0.07	−0.03	0.01
	Independence	−0.04	−0.01	0.03	0.13	−0.12	−0.05	0.02	0.03	0.13	−0.07	0.04	0.03	0.03	−0.05	−0.14	0.03	0.09	−0.08	-0.02
	Legality	0.09	0.04	0.10	0.15	−0.18	−0.09	0.12	0.04	0.13	−0.02	0.02	0.01	0.00	−0.03	−0.15	−0.09	0.08	−0.06	-0.07
	Maturity	0.12	0.18	0.16	**0.28**	**−0.33**	−0.13	**0.25**	0.07	**0.29**	0.08	0.18	−0.18	−0.09	−0.13	**−0.24**	−0.11	−0.07	**−0.22**	0.15
Subjective adult status		0.13	0.15	0.09	**0.20**	−0.19	−0.11	**0.25**	0.05	**0.21**	0.03	0.07	−0.11	−0.05	−0.11	−0.18	−0.08	−0.01	−0.17	0.03
		**ESTABLISHED ADULTHOOD**																		
		**WB**	**SL**	**OP**	**SE**	**AL**	**GHQ**	**SWB−SI**	**SWB−SA**	**SWB−SC**	**SWB−SL**	**SWB−SH**	**MAC**	**PSY**	**NAR**	**NA**	**DT**	**DL**	**DN**	**AN**
Imp	Total	0.03	0.18	**0.35**	0.14	−0.13	**−0.21**	**0.26**	0.07	0.20	+0.09	0.08	−0.01	0.00	0.15	−0.06	−0.15	0.08	−0.21	0.10
	Role	0.01	0.15	**0.28**	0.09	−0.02	−0.16	0.11	−0.05	0.06	−0.06	0.09	0.02	0.05	0.23	−0.03	−0.12	0.07	−0.10	0.00
	Independence	0.00	0.14	**0.23**	0.09	−0.07	−0.15	0.18	0.07	0.19	0.11	0.14	−0.04	0.01	0.10	−0.02	−0.07	0.05	−0.19	0.12
	Legality	−0.02	0.17	**0.34**	0.15	−0.12	−0.13	**0.30**	0.12	0.18	0.05	−0.05	−0.01	0.05	0.06	−0.14	−0.20	0.08	−0.15	0.04
	Maturity	0.12	0.08	**0.24**	0.12	**−0.30**	**−0.24**	**0.29**	0.11	**0.26**	**0.25**	0.07	−0.04	−0.18	0.05	−0.02	−0.11	0.06	**−0.29**	0.22
Ach	Total	**0.31**	**0.40**	**0.31**	**0.38**	−0.22	**−0.28**	**0.32**	**0.36**	**0.39**	0.19	0.13	0.02	−0.08	−0.10	−0.17	**−0.28**	−0.03	−0.10	0.02
	Role	**0.26**	**0.26**	0.17	**0.21**	−0.01	−0.14	0.17	0.17	**0.23**	0.02	−0.03	0.00	−0.10	−0.11	0.04	−0.15	−0.11	−0.04	0.06
	Independence	**0.27**	**0.33**	0.17	**0.27**	−0.20	−0.20	**0.26**	**0.34**	**0.34**	0.20	0.10	0.03	−0.05	−0.05	−0.17	**−0.28**	0.02	−0.09	-0.04
	Legality	0.16	**0.27**	**0.30**	**0.39**	**−0.34**	**−0.34**	**0.32**	**0.45**	**0.31**	0.20	**0.26**	−0.10	−0.02	−0.12	**−0.24**	**−0.26**	−0.08	−0.03	-0.07
	Maturity	0.18	**0.36**	**0.47**	**0.44**	**−0.26**	**−0.29**	**0.28**	0.17	**0.31**	**0.23**	**0.25**	0.07	−0.03	−0.02	**−0.28**	−0.11	0.12	−0.14	0.12
Subjective adult status		0.13	**0.30**	**0.21**	**0.23**	−0.20	**−0.31**	**0.40**	0.12	**0.25**	0.07	0.02	−0.14	0.01	−0.04	−0.20	−0.17	−0.10	−0.18	0.04
		**MIDLIFE**																		
		**WB**	**SL**	**OP**	**SE**	**AL**	**GHQ**	**SWB−SI**	**SWB−SA**	**SWB−SC**	**SWB−SL**	**SWB−SH**	**MAC**	**PSY**	**NAR**	**NA**	**DT**	**DL**	**DN**	**AN**
Imp	Total	−0.02	0.09	−0.01	0.02	−0.02	0.04	0.07	−0.10	0.10	−0.03	−0.03	0.03	−0.01	0.09	0.02	−0.09	0.10	−0.03	0.09
	Role	−0.04	0.09	0.02	0.00	0.07	0.03	0.05	−0.09	0.06	−0.07	−0.04	0.07	0.05	0.06	0.02	−0.01	0.12	0.02	0.08
	Independence	−0.01	0.08	−0.02	0.06	−0.05	0.02	0.03	−0.11	0.07	−0.02	−0.04	0.00	−0.03	0.08	0.00	−0.11	0.06	−0.05	0.07
	Legality	−0.04	0.03	−0.04	−0.03	0.02	0.06	0.03	−0.11	0.08	−0.01	−0.03	0.03	0.03	0.10	0.05	−0.07	0.10	0.01	0.05
	Maturity	0.07	0.04	0.00	0.03	−0.18	0.04	0.15	0.07	0.09	0.06	0.05	−0.03	−0.13	−0.02	−0.05	−0.12	−0.05	−0.11	0.07
Ach	Total	0.16	**0.30**	**0.22**	0.30	−0.22	−0.15	0.17	0.12	0.17	0.18	0.15	−0.04	−0.05	0.01	**−0.26**	−0.14	−0.03	−0.20	0.00
	Role	0.06	**0.25**	0.11	0.11	−0.08	−0.01	0.05	0.06	0.06	0.07	0.02	0.02	−0.05	0.01	−0.06	−0.10	−0.05	−0.14	0.02
	Independence	0.11	**0.21**	0.18	**0.26**	−0.12	−0.13	0.12	0.08	0.14	0.16	0.11	−0.03	0.02	0.06	**−0.21**	−0.09	0.01	−0.13	-0.01
	Legality	0.11	0.17	0.14	0.19	−0.18	−0.10	0.14	0.11	0.13	0.17	0.15	0.00	−0.05	0.07	−0.13	−0.09	0.03	−0.09	0.00
	Maturity	**0.21**	**0.20**	**0.20**	**0.28**	**−0.31**	−0.18	0.20	0.11	0.16	0.11	0.17	−0.12	−0.13	−0.13	**−0.31**	−0.15	−0.06	−0.18	0.00
Subjective adult status		0.17	0.13	0.13	**0.25**	−0.14	−0.17	0.18	0.06	0.15	0.11	0.10	−0.04	−0.04	0.04	−0.16	−0.09	−0.10	**−0.23**	0.08
		**OLDER ADULTHOOD**																		
		**WB**	**SL**	**OP**	**SE**	**AL**	**GHQ**	**SWB−SI**	**SWB−SA**	**SWB−SC**	**SWB−SL**	**SWB−SH**	**MAC**	**PSY**	**NAR**	**NA**	**DT**	**DL**	**DN**	**AN**
Imp	Total	−0.06	0.12	0.04	0.01	0.06	0.12	0.22	−0.05	0.11	−0.05	−0.10	−0.13	−0.06	0.04	0.08	−0.08	−0.11	−0.15	**0.24**
	Role	−0.05	0.16	0.03	0.00	0.10	0.08	0.11	−0.05	0.08	−0.07	−0.08	−0.07	0.03	0.11	0.07	−0.01	−0.04	−0.09	0.17
	Independence	−0.04	0.07	0.01	−0.02	0.05	0.14	0.22	−0.04	0.06	0.05	−0.08	−0.06	−0.09	0.05	0.06	−0.09	−0.14	−0.17	**0.22**
	Legality	−0.12	0.03	−0.01	−0.07	0.12	0.12	0.14	−0.10	0.07	−0.13	−0.13	−0.10	0.00	0.00	0.12	−0.02	0.01	−0.07	0.16
	Maturity	0.07	0.14	0.14	0.16	−0.15	0.00	**0.29**	0.03	**0.21**	0.00	0.00	−0.23	−0.18	−0.10	−0.02	−0.18	**−0.23**	−0.20	**0.23**
Ach	Total	0.16	**0.26**	**0.25**	**0.28**	**−0.21**	−0.11	**0.24**	0.14	**0.29**	0.16	0.17	−0.04	−0.03	0.06	−0.16	**−0.21**	0.04	−0.17	0.05
	Role	0.08	**0.21**	0.13	0.10	−0.06	0.03	0.13	0.08	0.06	0.07	0.04	0.02	0.07	−0.01	−0.03	−0.14	0.06	−0.03	0.01
	Independence	0.17	0.17	**0.21**	**0.27**	−0.18	−0.14	0.17	0.12	**0.30**	0.18	0.14	−0.06	−0.05	0.12	−0.14	−0.15	0.05	−0.15	0.07
	Legality	0.10	0.10	0.16	**0.22**	**−0.23**	−0.11	0.19	0.13	**0.25**	0.15	0.18	0.02	−0.04	0.02	−0.18	−0.17	0.01	−0.17	0.01
	Maturity	0.12	**0.27**	**0.21**	0.18	−0.16	−0.12	**0.22**	0.02	**0.21**	0.03	0.12	−0.13	−0.09	0.00	−0.14	−0.16	−0.04	−0.13	0.03
Subjective adult status		**0.24**	0.05	0.14	**0.23**	**−0.21**	−0.13	0.15	0.18	0.12	0.20	0.15	−0.14	−0.12	0.00	**−0.20**	−0.15	−0.12	**−0.21**	0.07

WB: well-being; SL: satisfaction with life; OP: optimism; SE: self-esteem; AL: Alexithymia; GHQ: General Health Questionnaire; SWB-SI: social well-being–social integration; SWB-SA: social well-being–social acceptance; SWB-SC: social well-being–social contribution; SWB–SL: social actualization; SWB-SH: social well-being–social coherence; MAC: Machiavellianism; PSY: Psychopathy; NAR: Narcissism; NA: negative affectivity; DT: detachment; DL: dissociality; DN: disinhibition; AN: anankastia. Correlations higher than +/−0.20 are in boldface. Correlations equal to or higher than ±0.08, 0.11, 0.30, 0.14, and 0.20 for the total sample, emerging adulthood, established adulthood, midlife, and older adulthood, respectively, had *p*-values lower than 0.001. Exact *p*-values for each correlation can be obtained upon request from the corresponding author.

**Table 2 ijerph-21-00773-t002:** Orthogonal pattern matrix of mental health indexes.

	Negative Emotions	Social Well-Being	Antisocial	Disinhibition
WB	**0.45**	**0.31**	−0.02	0.23
SL	**0.53**	0.27	0.03	0.17
OP	**0.70**	**0.31**	−0.02	0.02
SE	**0.81**	0.25	0.01	0.11
AL	**−0.49**	**−0.46**	0.18	−0.11
GHQ	**−0.72**	0.01	0.11	0.07
SWB-SI	**0.38**	**0.54**	0.10	0.17
SWB-SA	0.18	**0.55**	**−0.31**	−0.24
SWB-SC	**0.41**	**0.54**	0.03	0.19
SWB-SL	0.07	**0.68**	−0.11	−0.04
SWB-SH	0.26	**0.46**	0.03	−0.17
MAC	−0.10	−0.04	**0.82**	−0.07
PSY	0.15	**−0.41**	**0.62**	−0.17
NAR	−0.18	0.21	**0.73**	0.13
NA	**−0.77**	−0.09	0.25	0.20
DT	−0.15	**−0.72**	0.07	−0.02
DL	−0.03	−0.20	**0.77**	−0.17
DN	**−0.34**	−0.10	**0.41**	**−0.64**
AN	−0.04	−0.12	−0.04	**0.88**

WB: well-being; SL: satisfaction with life; OP: optimism; SE: self-esteem; AL: Alexithymia; GHQ: General Health Questionnaire; SWB-SI: social well-being–social integration; SWB-SA: social well-being–social acceptance; SWB-SC: social well-being–social contribution; SWB–SL: social actualization; SWB-SH: social well-being–social coherence; MAC: Machiavellianism; PSY: Psychopathy; NAR: Narcissism; NA: negative affectivity; DT: detachment; DL: dissociality; DN: disinhibition; AN: anankastia. Loadings higher than +/−0.30 are in boldface.

**Table 3 ijerph-21-00773-t003:** Correlations among perceptions of the importance (Imp) and achievement (Ach) of markers of adulthood and subjective adult status with factor scores in the total sample and the four age stages.

		Total Sample				Emerging Adulthood				Established Adulthood				Midlife				Older Adulthood			
		NE	SWB	ANT	DIS	NE	SWB	ANT	DIS	NE	SWB	ANT	DIS	NE	SWB	ANT	DIS	NE	SWB	ANT	DIS
Imp	Total	0.05	0.02	0.00	0.15	0.01	0.01	0.08	0.12	**0.23**	0.17	0.11	0.17	0.00	0.04	0.10	0.13	−0.05	0.04	−0.01	**0.29**
	Role	0.13	−0.04	−0.01	0.08	0.11	−0.05	0.11	0.01	0.18	0.05	0.15	0.08	0.02	−0.03	0.12	0.09	−0.01	−0.01	0.06	**0.21**
	Independence	0.00	0.01	0.02	0.14	−0.04	−0.03	0.07	0.12	0.15	0.14	0.07	0.15	0.01	0.03	0.07	0.13	−0.10	0.09	−0.03	**0.27**
	Legality	0.01	0.00	0.05	0.10	0.00	−0.01	0.09	0.08	**0.24**	0.14	0.09	0.08	−0.05	0.03	0.11	0.09	−0.08	−0.05	0.01	0.19
	Maturity	−0.02	0.18	−0.07	0.16	−0.08	0.18	−0.12	0.16	0.16	**0.25**	0.02	**0.27**	0.00	0.15	−0.05	0.10	0.09	0.14	−0.15	**0.27**
Ach	Total	**0.33**	0.06	**−0.32**	0.06	**0.21**	0.05	0.00	0.04	**0.38**	**0.34**	0.02	0.05	**0.29**	0.17	0.00	0.05	**0.25**	**0.26**	0.09	0.10
	Role	**0.26**	0.05	**−0.30**	0.07	0.05	0.09	0.02	0.05	0.17	0.15	−0.02	0.12	0.10	0.09	0.00	0.08	0.10	0.12	0.08	0.03
	Independence	**0.30**	0.03	**−0.31**	0.04	0.13	−0.04	0.02	−0.01	**0.28**	**0.33**	0.04	0.00	**0.24**	0.11	0.04	0.02	**0.22**	**0.24**	0.09	0.11
	Legality	**0.29**	0.09	−0.19	0.02	0.17	0.05	0.03	−0.02	**0.39**	**0.34**	−0.06	−0.13	0.16	0.17	0.05	0.02	0.19	**0.23**	0.04	0.04
	Maturity	**0.34**	0.13	−0.18	0.12	**0.26**	0.13	−0.10	0.17	**0.49**	**0.22**	0.12	0.09	**0.31**	0.13	−0.11	0.02	**0.22**	0.15	0.00	0.10
Subjective adult status		0.18	**0.31**	0.09	**−0.23**	**0.21**	0.09	−0.05	0.09	**0.35**	0.15	−0.03	0.11	**0.22**	0.11	−0.01	0.13	0.17	**0.21**	−0.10	0.08

NE: negative emotions; SWB: subjective well-being; ANT: antisocial; DIS: disinhibition. Correlations larger than +/−0.20 are in boldface. Correlations equal to or higher than ±0.08 had a *p*-value lower than 0.001. Exact *p*-values for each correlation can be obtained upon request from the corresponding author.

**Table 4 ijerph-21-00773-t004:** Standardized regression coefficients of the final equation of the hierarchical regression analyses and its change in the R^2^ for the total sample and the four age stages.

		Total Sample															
		Negative Emotions				Social Well-Being				Antisocial				Disinhibition			
		*b*	t	sig	Change R^2^	*b*	t	p	Change R^2^	*b*	t	sig	Change R^2^	*b*	t	sig	Change R^2^
	Age	−0.07	−1.46	0.15	0.053	−0.10	−2.12	0.03	0.039	**−0.34**	−7.40	0.00	0.126	−0.03	−0.70	0.49	0.003
	SPI	−0.08	−2.94	0.00		**−0.24**	−8.13	0.00		−0.09	−3.21	0.00		−0.05	−1.48	0.14	
IMP	Role	0.12	4.31	0.00	0.007	−0.06	−2.06	0.04	0.027	0.10	3.55	0.00	0.026	−0.01	−0.34	0.73	0.034
	Independence	−0.03	−1.05	0.30		−0.01	−0.33	0.74		0.02	0.61	0.54		0.10	3.33	0.00	
	Legality	−0.04	−1.48	0.14		0.00	−0.15	0.88		0.05	1.75	0.08		0.02	0.71	0.48	
	Maturity	−0.08	−3.21	0.00		0.14	5.83	0.00		−0.11	−4.80	0.00		0.10	4.06	0.00	
ACH	Role	−0.09	−2.00	0.05	0.121	0.17	3.51	0.00	0.035	−0.04	−0.89	0.37	0.005	0.13	2.58	0.01	0.012
	Independence	**0.29**	5.20	0.00		0.05	0.79	0.43		0.04	0.66	0.51		−0.02	−0.27	0.79	
	Legality	0.11	3.59	0.00		0.08	2.40	0.02		0.03	1.05	0.29		−0.05	−1.41	0.16	
	Maturity	**0.27**	11.42	0.00		0.08	3.27	0.00		−0.07	−2.93	0.00		0.09	3.51	0.00	
		**EMERGING ADULTHOOD**															
		**Negative Emotions**				**Social Well-Being**				**Antisocial**				**Disinhibition**			
		** *b* **	**t**	**sig**	**Change R^2^**	** *b* **	**t**	**sig**	**Change R^2^**	** *b* **	**t**	**sig**	**Change R^2^**	** *b* **	**t**	**sig**	**Change R^2^**
	Age	0.06	1.21	0.23	0.022	−0.02	−0.31	0.76	0.025	−0.12	−2.29	0.02	0.002	0.00	0.00	1.00	0.008
	SPI	0.04	0.98	0.32		−0.16	−3.96	0.00		0.02	0.43	0.67		−0.09	−2.15	0.03	
IMP	Role	0.15	3.81	0.00	0.020	−0.07	−1.62	0.11	0.038	0.08	2.00	0.05	0.036	−0.06	−1.38	0.17	0.030
	Independence	−0.06	−1.56	0.12		−0.06	−1.56	0.12		0.05	1.23	0.22		0.09	2.16	0.03	
	Legality	0.00	−0.03	0.97		0.00	0.09	0.93		0.05	1.19	0.24		0.04	1.06	0.29	
	Maturity	−0.11	−3.15	0.00		0.17	4.69	0.00		−0.13	−3.59	0.00		0.09	2.51	0.01	
ACH	Role	−0.09	−2.44	0.01	0.083	0.09	2.16	0.03	0.023	0.01	0.20	0.84	0.011	0.05	1.22	0.22	0.026
	Independence	0.02	0.36	0.72		−0.02	−0.45	0.65		0.07	1.33	0.18		0.03	0.53	0.59	
	Legality	0.09	2.21	0.03		0.08	1.92	0.06		0.06	1.34	0.18		−0.04	−0.90	0.37	
	Maturity	**0.28**	7.96	0.00		0.08	2.34	0.02		−0.09	−2.53	0.01		0.15	4.05	0.00	
		**ESTABLISHED ADULTHOOD**															
		**Negative Emotions**				**Social Well-Being**				**Antisocial**				**Disinhibition**			
		** *b* **	**t**	**sig**	**Change R^2^**	** *b* **	**t**	**sig**	**Change R^2^**	** *b* **	**t**	**sig**	**Change R^2^**	** *b* **	**t**	**sig**	**Change R^2^**
	Age	−0.01	−0.10	0.92	0.022	−0.04	−0.47	0.64	0.034	0.08	0.77	0.44	0.013	0.05	0.52	0.60	0.010
	SPI	−0.01	−0.14	0.89		−0.07	−0.76	0.45		−0.12	−1.20	0.23		−0.01	−0.11	0.91	
IMP	Role	0.18	1.65	0.10	0.055	0.14	1.12	0.26	0.049	0.15	1.10	0.27	0.025	−0.08	−0.65	0.52	0.086
	Independence	−0.02	−0.19	0.85		−0.09	−0.73	0.46		−0.02	−0.16	0.88		0.06	0.44	0.66	
	Legality	0.04	0.36	0.72		−0.08	−0.61	0.54		0.04	0.29	0.77		−0.08	−0.60	0.55	
	Maturity	0.03	0.24	0.81		**0.28**	2.44	0.02		−0.09	−0.67	0.50		**0.35**	2.91	0.00	
ACH	Role	−0.04	−0.36	0.72	0.266	−0.01	−0.10	0.92	0.157	−0.14	−1.04	0.30	0.030	**0.29**	2.28	0.02	0.075
	Independence	0.12	1.07	0.29		**0.29**	2.46	0.02		0.12	0.89	0.38		−0.12	−0.96	0.34	
	Legality	**0.26**	2.72	0.01		**0.23**	2.20	0.03		−0.12	−1.08	0.28		**−0.23**	−2.11	0.04	
	Maturity	**0.35**	3.95	0.00		0.01	0.11	0.91		0.13	1.22	0.23		0.11	1.10	0.28	
		**MIDLIFE**															
		**Negative Emotions**				**Social Well-Being**				**Antisocial**				**Disinhibition**			
		** *b* **	**t**	**sig**	**Change R^2^**	** *b* **	**t**	**sig**	**Change R^2^**	** *b* **	**t**	**sig**	**Change R^2^**	** *b* **	**t**	**sig**	**Change R^2^**
	Age	0.05	1.23	0.22	0.031	0.03	0.86	0.39	0.069	−0.01	−0.18	0.86	0.016	−0.02	−0.55	0.58	0.003
	SPI	−0.12	−3.12	0.00		−0.22	−5.51	0.00		−0.13	−3.16	0.00		−0.04	−0.99	0.32	
IMP	Role	0.06	1.29	0.20	0.007	−0.08	−1.55	0.12	0.019	0.11	2.12	0.03	0.026	0.02	0.43	0.67	0.021
	Independence	0.06	1.12	0.26		0.02	0.49	0.62		0.00	0.00	1.00		0.10	1.82	0.07	
	Legality	−0.10	−2.00	0.05		0.05	0.96	0.34		0.07	1.33	0.19		0.01	0.20	0.84	
	Maturity	−0.07	−1.79	0.07		0.11	2.62	0.01		−0.07	−1.64	0.10		0.06	1.41	0.16	
ACH	Role	−0.03	−0.63	0.53	0.110	0.07	1.53	0.13	0.018	−0.02	−0.55	0.59	0.018	0.09	1.98	0.05	0.006
	Independence	0.15	3.36	0.00		−0.01	−0.16	0.87		0.04	0.88	0.38		−0.04	−0.72	0.47	
	Legality	−0.01	−0.23	0.82		0.07	1.64	0.10		0.06	1.28	0.20		0.01	0.19	0.85	
	Maturity	**0.28**	6.84	0.00		0.06	1.47	0.14		−0.13	−3.18	0.00		0.00	−0.02	0.99	
		**OLDER ADULTHOOD**															
		**Negative Emotions**				**Social Well-Being**				**Antisocial**				**Disinhibition**			
		** *b* **	**t**	**sig**	**Change R^2^**	** *b* **	**t**	**sig**	**Change R^2^**	** *b* **	**t**	**p**	**Change R^2^**	** *b* **	**t**	**p**	**Change R^2^**
	Age	−0.10	−1.52	0.13	0.080	−0.07	−1.08	0.28	0.078	−0.09	−1.38	0.17	0.002	0.02	0.26	0.80	0.002
	SPI	**−0.23**	−3.40	0.00		−0.16	−2.29	0.02		−0.04	−0.57	0.57		0.02	0.24	0.81	
IMP	Role	0.12	1.52	0.13	0.035	0.04	0.47	0.64	0.029	0.13	1.51	0.13	0.047	0.12	1.47	0.14	0.105
	Independence	**−0.22**	−2.43	0.02		0.13	1.45	0.15		−0.04	−0.44	0.66		0.15	1.58	0.12	
	Legality	−0.09	−0.97	0.33		**−0.24**	−2.66	0.01		0.06	0.65	0.52		−0.05	−0.58	0.57	
	Maturity	0.07	0.90	0.37		0.08	0.96	0.34		**−0.24**	−2.88	0.00		0.17	2.15	0.03	
ACH	Role	0.06	0.85	0.40	0.059	0.05	0.79	0.43	0.041	0.05	0.64	0.52	0.012	−0.05	−0.65	0.52	0.005
	Independence	0.14	1.76	0.08		0.07	0.85	0.40		0.09	1.05	0.30		0.06	0.69	0.49	
	Legality	−0.03	−0.35	0.73		0.11	1.39	0.17		−0.02	−0.26	0.79		0.00	0.04	0.97	
	Maturity	0.16	2.43	0.02		0.09	1.30	0.20		0.03	0.41	0.68		0.04	0.56	0.58	

Standardized beta coefficients higher than +/−0.20 in boldface.

## Data Availability

The raw data supporting the conclusions of this article will be made available by the authors on request.

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
