# Peer review of "Perceptions of Adulthood and Mental Health"

_ijerph, 2024, doi:10.3390/ijerph21060773_

Round 1
Reviewer 1 Report
Comments and Suggestions for Authors
This study applied some adulthood markers (Markers of Adulthood scale (MoA), Subjective adult status), and multiple mental health indexes (Well-Being (WB), Satisfaction with Life (SL), Optimism (OP), Self-Esteem (SE), Alexithymia (AL), General Health Questionnaire (GHQ), Social Well-being, Dark Triad (The Dirty Dozen), Dimensional Personality Disorders (The Personality Inventory for ICD-11 – PiCD) to a community sample comprising 1,772 individuals (754 males and 1,018 females) from the Spanish general population, ages from 16 to 93 years. Authors conclude that perceptions of adulthood display strong correlations with nearly every assessed index of mental health, particularly those that comprise a dimension of negative emotions.
Advantages
• Manuscript represents a contribution to the yet poorly researched field
• According to authors this is the pionir study examining the complex interactions
• Use of large set of instruments
• The results might have a potential application in terms of preventive public health approaches
Weackness
• Preliminary results and further research is needed
• Subjects - uneven distribution in individual age groups (e.g. in the age group of 30-45 years), social positions (e. g. low-middle)
Title is clear and communicative. The abstract is appropriate. The introduction is coherent and adequately lead to the objectives. The methods are appropriate and with comprehensive description and accuracy of methods. The findings are appropriately discussed. The discussion is well written and study limitations are reported. In my opinion, this is an interesting paper that could provide an useful references for many researchers in the field.
This scholarly written piece of work deserves publication. I suggest accepting the study as it is.
Author Response
We really appreciate the generally positive view expressed by the reviewer which encourage us to continue. Note that some amendments have been included to reduce or discuss the weaknesses of the manuscript.
Reviewer 2 Report
Comments and Suggestions for Authors
The "Perceptions of Adulthood and Mental Health" manuscript covers a crucial aspect of human life. There are some important and strong points in the current manuscripts. However, I am also proposing some additions to enhance its usefulness.
1. Its novel idea of understanding the relationship between perceptions of adulthood and mental health is crucial, as many mental health aspects become evident at this age. The current manuscript can cover some specific mental health issues that have been researched in this specific stage (mentioning categories of phases included by authors, like late adulthood, etc.) in geenral or specifc to spanish culture.
2. The data is collected on the Spanish population and paper also covers the importance of cultural context on the perception of adulthood. Here, authors can include some specific aspects of this culture to present the audience with a comparison of their cultural. Obviously, in brief
2. The scales chosen here are diverse and good. The authors have used capital or small capslocks in different places differently that can be recheked once.
4. In the discussion version, a theoretical link from Section 1.1 can be used to connect it rationale given by the authors.
5. For a small correction, I highlighted a place on page one.
In its discussion and implication section, the current paper explores specific criteria for various generations throughout adulthood. However, this information is not included in the results section. It would be more appropriate if various stages of adulthood could be discussed in detail in the discussion and implication section. Otherwise, incorporating these stages will not serve its purpose.
7. Discussion sections 410–415 present significant findings, but one can use a solid theoretical understanding to comprehend or validate these findings. The same applies to other important findings about negative emotions.
8. A grammar recheck will be helpful.
I appreciate the author's novel and relevant work. I wish the best for all authors.

Author Response
We really appreciate the positive view about the article expressed by this reviewer. We also think that his/her comments help us to improve the present draft of the manuscript. The first thing we want to highlight is that English style has been reviewed again by the first author that is completely fluent in English as well as by a native-speaker. Both of them agree that the manuscript presents a proper English-style. Note that some changes have explicitly been included to improve this formal aspect. However, if the reviewer still feels that some parts of the manuscript present some problems in regard to the language, we will really appreciate that you to point out some examples.
Regarding your comments, we state below the amendments conducted according to your suggestions.
- We agree that the own characteristics of the Spanish society could affect to Perceptions of adulthood and to the results of the research line introduced in the paper. So, we have added to the discussion section some specific characteristics of the Spanish adulthood suggested by other authors to be considered in future research, especially in cross-cultural research. So, those specific aspects of the country have been added to the future studies section as follows: “To address the variability in adulthood perceptions across cultures [2, 3, 4, 10], future research should aim to replicate our study in countries with different sociological frameworks, particularly in non-Western countries. This expansion would provide valuable insights into the cultural nuances of adulthood and how these differences in-fluence mental health outcomes. Cultural norms, values, and expectations significantly shape the transition to adulthood, making it imperative to examine these constructs in varied cultural settings. For instance, it has been described that Spanish people em-phasize psychological aspects (emotional and sexual) over financial or chronological criteria in the transition to adulthood compared to other countries [10]. They are also characterized by a longer stay in the parental home, higher rates of enrollment in higher education, delayed workforce entry, and older marriage ages. Different results could be found in countries with different sociological profiles. Alongside cultural diversity, the inclusion of diverse social groups within the same cultural context is crucial. This ap-proach allows for an examination of how socio-economic status, education levels, and other social determinants impact perceptions of adulthood and their association with mental health outcomes. Conducting comparative analyses between Western and non-Western countries, as well as across different social groups within the same coun-try, would shed light on the relative impact of cultural and social factors on adulthood perceptions. Such analyses could highlight universal versus context-specific elements of the transition to adulthood and its relationship with mental health. Lastly, considering the somewhat low reliability of the subjective adult status scale observed in the Spanish context, it is important to replicate its psychometric properties in other countries to support the universal applicability of this promising scale”. (p. 19).
- We have checked the manuscript to detect an inappropriate use of capital or small capslocks.
- We agree with reviewer that some connection between introduction and discussion sections is missing. So, the following paragraphs of the discussion section have been rewritten and extended: “The present manuscript supports the notion that an individual's perception of being integrated and productive within society plays a significant role in their mental health. In detail, perceptions of adulthood account for about 10% of the reliable variance in different mental health measures, highlighting their significance in addressing individual and societal mental health differences. This insight is particularly valuable in psychological research aimed at pinpointing and comprehending the factors influencing mental health, which is essential for creating effective interventions and support mechanisms. The finding that achievement scales, rather than importance scales, predict mental health outcomes suggests that the actual perception of achieving adulthood milestones, particularly psychological ones, is most beneficial. It is not the perceived importance of these elements, but rather the perception of having attained them, that positively influences mental health. This distinction highlights the practical implications of adulthood achievements, suggesting that the perception of realization of key developmental tasks is more impactful on an individual's mental health than simply valuing these milestones. The pivotal role of achieving relational maturity as the most predictive factor for the connection between perceptions of adulthood and mental health underscores a significant finding in this study, replicating previous literature [13, 27]. This suggests that a mature understanding and management of one's abilities, alongside a willingness to accept the consequences of one's actions, are crucial elements linking adulthood perceptions to mental health outcomes. These insights highlight the critical role of psychological maturity in adulthood and mental health well-being, reinforcing the importance of this concept as outlined in reference [20]. (p. 17).
- We also think that more information about the age stages in the results section is lack. So, some paragraphs have been extended as follows: “Table 3 reveals stronger correlations between achievement scales, subjective adult status, and especially the Negative Emotions factor, with a lesser correlation observed with the Antisocial factor. Associations with Negative Emotions remain consistent across all age stages, while Social Well-being correlates with achievement scales during established adulthood and midlife, and the Antisocial factor correlates with achievement scales in the total sample but not within specific age stages.
Table 4 demonstrates that achievement scales serve as key predictors in hierarchical regressions, contributing up to an additional 12% of the variance in the Negative Emotions factor for the total sample, after accounting for age, SPI, and achievement scales. In the four age stages, this increment ranges from 6% in older adulthood to 26% in established adulthood. The relative maturity scale emerges as the strongest predictor of Negative Emotions, consistently showing the low predictive power of age, SPI, and importance scales across age stages. Notably, the percentage of variance explained by the other three factors is lower, particularly for the Antisocial and Disinhibition factors.” (p. 12).
In regard to this point, note that the results observed for different age stages are commented in different paragraphs of the discussion section as, for instance, in those two paragraphs: “Interestingly, the variance in the strength of associations across the four adulthood stages indicates that the transition to and experience of adulthood are not monolithic but vary significantly with age. This insight introduces a refined perspective on adulthood, emphasizing the varying impact it has across different life stages. The discovery that perceptions of adulthood are most predictive during established adulthood (ages 30-45) suggests that this stage's unique social and psychological pressures heavily in-fluence mental health. This stage of adulthood is often marked by intensified demands from various aspects of personal and professional life, which may amplify the impact of adulthood perceptions on mental health.” (p. 17) and “Finally, the present paper recommends redirecting attention towards psychosocial interventions tailored specifically for individuals in the established adulthood stage. Customized programs aimed at assisting adults in managing the intricacies and obstacles of this stage, including aspects such as work-life balance, parenting, and career advancement, are crucial. Support groups, counseling services, and workplace initiatives can all play significant roles in mitigating the mental health risks associated with this life stage.” (p. 20).
Finally, we would like to emphasize that there still is not a solid theoretical framework of this research line about perceptions of adulthood. As it has been stated in some places along the paper, this is a novel work that pretends to initiate a fruitful research line. Note that we have put a lot of emphasis on future studies to open avenues of inquiry that should be developed in the decades to come.
Reviewer 3 Report
Comments and Suggestions for Authors
This manuscript describes a cross sectional study which analyzed the association between the perception of adulthood milestones and general indicators of positive mental health and psychopathology. The utilization of the construct perception of adulthood milestones seems very interesting, in particular, when we consider such construct as a proxy measurement of developmental adjustment. Regardless of the relevance of the construct, I find that this work has several limitations, which I detail below.
1. Title: The title is too wide; it is unclear the characteristics of the study. It requires to be more specific.
2. In the introduction several disorders are mentioned (page 3, lines 118-121) however only personality disorders were measured.
3. Several variables related to mental health were included, however, the theoretical rationale for their selection is unclear. It this sense is important to note that there should be a clear rationale to include each variable, also if there is some sort of underlying theoretical framework, it would be useful to include it, otherwise it only seems that the selection of variables was arbitrations.
4. The study objective is not clearly described. Please describe the associations that are intended to test as part of you study. While the last paragraph mention that there is the primary objective was to test “this association” is hard to read it as it is.
Methods:
5. How was sample size calculated? How was estimated the size of every stratum?
6. It seems that the instruments that measure perceptions of adulthood haven been assess in Spanish population. Do you have any information regarding their validity, in addition to their alphas? If the validity of these variables is unclear, the validity of the overall results is jeopardized.
7. Please add the information about the IRB or ethics committee review and approval in the procedure.
8. I find several other limitations in the analytic approach, however, the previously mentioned ethical limitations should be resolved first.
Author Response
We really appreciate the valuable amendments suggested by the reviewer. We have conducted many changes as it is described below. In some cases, we have not followed the amendment. Naturally, we expose our reasons for not doing so.
- Title: The title is too wide; it is unclear the characteristics of the study. It requires to be more specific.
- We understand that the title could be seem too general, but we have decided not to change it because we have tested different titles and all of them would restrict the message of the paper. Note that we have intended to test the relationships between perceptions of adulthood and a wide number of mental health dimensions. As far as we know, this is the first paper analyzing the association among perceptions of adulthood and a wide approach to mental health. So, we think that the title fits well to the manuscript. We also thought to introduce the term Spanish (or another specific characteristic of the sample and procedure) but we think that it would sound somewhat restrictive as well. Finally, we decided to retain the title because another reviewer agree explicitly with it. We hope reviewer understand our reasons.
- In the introduction several disorders are mentioned (page 3, lines 118-121) however only personality disorders were measured.
- You are right. We have included as many mental health indexes as possible, but it is quite impossible to introduce the extremely wide of disorders and other mental health aspects. Note that we are aware that many disorders are not considered in the present study so two paragraphs of the discussion section about future studies are focused on that point as follows: “The absence of direct measures of anxiety, depression, and other internalizing disorders in the current study highlights an essential area for future research. Considering the centrality of these concepts in any comprehensive mental health model [51], their inclusion could significantly enhance our understanding of the nuanced relationship between perceptions of adulthood and mental health outcomes. Previous studies showed a linkage between perceptions of adulthood and anxiety and depression [27, 56] that is theoretically consistent with our findings related to the Negative Emotions factor. Given that this factor encompasses aspects theoretically aligned with anxiety and mood disorders [57], integrating direct measures of these conditions could provide a more detailed understanding of the psychological challenges associated with the transition to and experiences of adulthood.
Moreover, broadening the scope of this research to encompass other aspects of mental health, such as substance abuse, could provide a more comprehensive under-standing of the challenges and vulnerabilities associated with adulthood. For instance, substance abuse, often linked to impulsivity or coping mechanisms for dealing with stress, anxiety, and depression [57], could further elucidate the complex interplay be-tween mental health and the societal and psychological aspects of adulthood. Incorporating these additional mental health indexes in future studies would not only validate and extend the findings of the present research but also contribute to a deeper and more comprehensive understanding of adulthood's psychological impact.” (p. 18).
- Several variables related to mental health were included, however, the theoretical rationale for their selection is unclear. It this sense is important to note that there should be a clear rationale to include each variable, also if there is some sort of underlying theoretical framework, it would be useful to include it, otherwise it only seems that the selection of variables was arbitrations.
- Author is also right when they state that there is not an unified theoretical rationale to include such mental health indexes. We have intended to measure several distinct dimensions of mental health including the most classical measures (as for instance, Well-being, Satisfaction with life or Alexithimia), social aspects, health, and personality pathological (Dark Triad and dimensional personality disorders). In fact, factor analysis was conducted to offer a more theoretical-sound approach to the mental health in the paper. The four-factor structure reported offers a quite comprehensive view of mental health and, above all, demonstrate that we have been somewhat successful adding different mental health indexes. In any case, and considering your righty comment, we have reinforced the message including specifically in the aim that different aspects of the mental health have been considered. So, the last paragraph of the introduction section was modified as follows: “The objective of this study is to explore the association between perceptions of adulthood and various dimensions of mental health. By extending the existing body of literature, this study aims to highlight the potential role of adulthood perceptions in enhancing our understanding of mental health. If significant associations are discovered, it would suggest that fostering a perception of adulthood and achieving adult status could positively impact the mental health of individuals, communities, and society at large.”. (p. 3)
- The study objective is not clearly described. Please describe the associations that are intended to test as part of you study. While the last paragraph mention that there is the primary objective was to test “this association” is hard to read it as it is.
- Note that this is the first study addressed to investigate this topic. As far as we know, there is only two previous studies, and both consider mainly a measure of well-being only. So, the aim is somewhat prelaminar and is addressed mostly to describe the possible associations among perceptions of adulthood and several mental health indexes. We have reinforced this idea in the paragraph commented in the previous point as well as in other places of the article as, for instance, this sentence on the discussion section: “As a groundbreaking study in this field, it lays the foundation for numerous future research avenues” (p. 18).
Methods:
- How was sample size calculated? How was estimated the size of every stratum?
- No previous sample size estimation was conducted. We intended to get as much sample and representative as possible to conduct multivariate analysis properly and could generalize conclusions to population.
- It seems that the instruments that measure perceptions of adulthood haven been assess in Spanish population. Do you have any information regarding their validity, in addition to their alphas? If the validity of these variables is unclear, the validity of the overall results is jeopardized.
- A manuscript about the Spanish adaptation of the markers of adulthood is currently under preparation. Since we already have the final results, we have included more information about them as well as the preliminary reference as follows: “The results of the adaptation process to the Spanish language and cultural context demonstrate adequate reliability for the scale, with alpha coefficients of .87 and .95 for the importance and achievement overall scores, respectively. Cronbach’s alpha values ranged from .67 to .85 for the importance scales and from .64 to .96 for the achievement scales. A similar factor structure to the original American version was also found [40].” (p. 4).
In regard to the Subjective adult status scale, more information has been added to describe the procedure and initial results of the adaptation process. So, the description of this last scale has been extended with the following sentences: “To adapt the scale to the Spanish language, the items were first translated by a Spanish psychology lecturer fluent in English. Subsequently, another Spanish psychology lecturer, also fluent in English, performed a back-translation. The back-translated version matched perfectly with the original English version. In the Spanish context, the reliability was indicated by a Cronbach’s alpha of .64. This coefficient could increase to .77 (closer to the original version's .82 [24]) if item 2 ("I no longer feel like a child") were removed, and to .84 if item 4 ("Other people treat me like a child") were also excluded. However, the original five-item composition has been retained for comparative purposes.” (p. 5).
- Please add the information about the IRB or ethics committee review and approval in the procedure.
- Note that all information about ethics committee is on the Institutional Review Board Statement already included in the article: “Institutional Review Board Statement: The study was conducted in accordance with the Declaration of Helsinki, and approved by the Institutional Review Board (or Ethics Committee) of Universidad Autónoma de Madrid (protocol codes CEI-108-2096 and CEI-123-2506) and Universidad de Lleida (CEIC 2160).”
- I find several other limitations in the analytic approach, however, the previously mentioned ethical limitations should be resolved first.
We hope that we had properly answered to the previous points, and we will wait for your comments about the analytic approach.
Reviewer 4 Report
Comments and Suggestions for Authors
The abstract should include the aim of the study.
The introduction is relevant and clear.
In the method, please explain why you opted for Hollingshead's Social Position Index two-factor instead of the four-factor version?
Some instruments mentioned that versions adapted to Spanish were used. How was the adaptation process to Spanish for those instruments that do not mention their Spanish version?
A low Cronbach's Alpha could have certain effects on markers of adulthood in the study. Please discuss the implications of this in your analysis.
Provide the gender distribution for all four age groups.
In the method section, please describe how the participants were invited to participate in the study. Also, provide a clear explanation of how the instruments were applied.
In the results section, it is necessary to include the p value in the tables.
In the conclusion section, please mention the most relevant findings of the study. Ensure that the conclusiones are directly realated to the findings.
Author Response
We appreciate the right amendments suggested by this reviewer. Below, you can find the changes conducted on the manuscript according to the different points.
The abstract should include the aim of the study.
- We agree with this importance absence. We have added the following sentence to the background section of the abstract: “The aim of this study is to explore the association between individuals' perceptions of adulthood and multiple dimensions of mental health.”
The introduction is relevant and clear.
- Thank you very much for your positive impression about the introduction.
In the method, please explain why you opted for Hollingshead's Social Position Index two-factor instead of the four-factor version?
- Since we did not gather necessary information to use a four-factor version, we used the two-factor one. However, we think is necessary to clarify this point, so we have added the following sentence: “Note that some necessary information, such as marital status, required to compute the four-factor version of the SPI was not collected; therefore, we used the two-factor version. (p. 4).
Some instruments mentioned that versions adapted to Spanish were used. How was the adaptation process to Spanish for those instruments that do not mention their Spanish version?
- A manuscript about the Spanish adaptation of the markers of adulthood is currently under preparation. Since we already have the final results, we have included more information about them as well as the preliminary reference as follows: “The results of the adaptation process to the Spanish language and cultural context demonstrate adequate reliability for the scale, with alpha coefficients of .87 and .95 for the importance and achievement overall scores, respectively. Cronbach’s alpha values ranged from .67 to .85 for the importance scales and from .64 to .96 for the achievement scales. A similar factor structure to the original American version was also found [40].” (p. 4).
In regard to the Subjective adult status scale, more information has been added to describe the procedure and initial results of the adaptation process. So, the description of this last scale has been extended with the following sentences: “To adapt the scale to the Spanish language, the items were first translated by a Spanish psychology lecturer fluent in English. Subsequently, another Spanish psychology lecturer, also fluent in English, performed a back-translation. The back-translated version matched perfectly with the original English version. In the Spanish context, the reliability was indicated by a Cronbach’s alpha of .64. This coefficient could increase to .77 (closer to the original version's .82 [24]) if item 2 ("I no longer feel like a child") were removed, and to .84 if item 4 ("Other people treat me like a child") were also excluded. However, the original five-item composition has been retained for comparative purposes.” (p. 5).
A low Cronbach's Alpha could have certain effects on markers of adulthood in the study. Please discuss the implications of this in your analysis.
- Note that low reliabilities (below .70) were observed for the subjective adult status scale only, and for some scale of the markers of adulthood. It should be remarked that total importance and achievement scales of this instrument, most of the scales and all mental health indexes presents Cronbach’s alpha higher than .70, and, in many cases, higher than .80. So, we think that the present manuscript is not affected by low reliability bias. However, since the reliability coefficient reported for the subjective adult status scale is lower than the original one, we have included a sentence on the discussion section about this point: “Lastly, considering the somewhat low reliability of the subjective adult status scale observed in the Spanish context, it is important to replicate its psychometric properties in other countries to support the universal applicability of this promising scale.” (p. 19).
Provide the gender distribution for all four age groups.
- We have added this information to the participants section as follows: “Consistent with prior literature [24], the sample was categorized into four age groups to align with the developmental stages of adulthood: (1) emerging adulthood (aged 18-29 years), which included 788 participants (44.5% of the sample; 290 males [36.8%] and 498 females [63.2%]); (2) established adulthood (aged 30-45 years), which comprised 116 participants (6.5%; 57 males [49.1%] and 59 females [50.9%]); (3) midlife (aged 46-59 years), which included 623 participants (35.2%; 296 males [47.5%] and 327 females [52.5%]); and (4) older adulthood (aged 60 years and above), which included 245 participants (13.8%; 111 males [45.3%] and 134 females [54.7%]).” (p. 3).
In the method section, please describe how the participants were invited to participate in the study. Also, provide a clear explanation of how the instruments were applied.
- We have added more information to clarify the application procedure. So, the following paragraph has been extended: “Undergraduate psychology students received training in the application of psychological assessment tools. As part of a routine exercise, they were tasked with ad-ministering a set of psychological measures in a paper-pencil format, including those discussed in this study, to seven individuals: students themselves, one male and one female aged between 18 and 30 years, one male and one female aged between 31 and 50 years, and one male and one female over the age of 51 years. This participant selection strategy was designed to capture a wide range of ages and genders. To ensure confidentiality, no names, personal identification numbers (such as identity cards), or any other personal details were recorded. However, all participants were informed about the study guidelines approved by the university's ethical committees. (p. 6).
In the results section, it is necessary to include the p value in the tables.
- We understand that the p values are a necessary information to interpret a correlation. However, it would be hard to include this information for all correlations. Besides, given the large sample size, many correlations are significant (even considering a very low p), so this information would not help to the reader to see interesting patterns. In any case, we think that it is necessary to point out what p values are associated to a relevant correlation. We have selected a quite restrictive cut-off p value (< .001) given the large sample size of the total sample. So, we have added this information to the footnote of tables 1: “Correlations equal to or higher than ±.08, .11, .30, .14, and .20 for the total sample, emerging adulthood, established adulthood, midlife, and older adulthood, respectively, had p-values lower than .001. Exact p-values for each correlation can be obtained upon request from the corresponding author.” (p. 10), and table 3: “Correlations equal to or higher than ±.08 had a p-value lower than .001. Exact p-values for each correlation can be obtained upon request from the corresponding author.” (p. 13).
In the conclusion section, please mention the most relevant findings of the study. Ensure that the conclusions are directly related to the findings.
- We appreciate your suggestion of summarizing the main findings of the paper in the conclusion section and connecting them with the aims of the introduction. It was a mistake not to have included it earlier. So, we have added a new first paragraph on the conclusion section. “The present study supports the association between perceptions of adulthood and mental health, particularly in relation to the risk of experiencing negative emotions. The findings also suggest that psychological aspects of adulthood are key to this association, opening an intriguing framework for the prevention and treatment of mental health issues.” (p. 20).
Besides, the structure of this section has been modified to adapt to the inclusion of this paragraph. So, we have also rewritten the further paragraphs. “The association varies across different adult stages. Developing strategies specifically targeted at each age group can help mitigate the onset and progression of mental health issues, emphasizing the importance of early detection and intervention. The high prevalence of mental health issues among adolescents and young adults underscores the significance of focusing on these age stages as crucial periods for the onset of psychiatric disorders [66]. Additionally, the majority of psychiatric disorders manifest during these stages [28]. The transition to adulthood is a critical time for fostering positive growth and integration into society, providing a stable and nurturing environment for individuals to thrive as healthy, contributing members of the community.
Understanding the risk and protective factors during these stages is essential for preventing mental health problems and improving treatment outcomes. Therefore, altering erroneous and maladaptive perceptions of adulthood and reinforcing subjective adult status could serve as valuable psychological tools to address these issues, potentially leading to long-term positive effects on mental health. The paradox concerning aging and subjective well-being, as discussed by Hansen & Blekesaune [67], suggests that despite the physical and social losses often associated with aging, subjective well-being tends to remain stable or even improve. This paradox could be partially at-tributed to enhanced perceptions of adulthood and social status serving as compensatory factors for the losses experienced during aging. Such perceptions may not only bolster well-being in later life but also potentially contribute to longevity [68].” (p. 20).
Round 2
Reviewer 4 Report
Comments and Suggestions for Authors
The suggestions were addressed correctly by the authors.
I only identified an error in reference 40, since the reference of the previous version was not deleted.